# *Orthoparamyxovirinae* C Proteins Have a Common Origin and a Common Structural Organization

**DOI:** 10.3390/biom13030455

**Published:** 2023-03-01

**Authors:** Ada Roy, Emeric Chan Mine, Lorenzo Gaifas, Cédric Leyrat, Valentina A. Volchkova, Florence Baudin, Luis Martinez-Gil, Viktor E. Volchkov, David G. Karlin, Jean-Marie Bourhis, Marc Jamin

**Affiliations:** 1Institut de Biologie Structurale, Université Grenoble Alpes, CNRS, CEA, 38000 Grenoble, France; 2Molecular Basis of Viral Pathogenicity, Centre International de Recherche en Infectiologie (CIRI), INSERMU1111-CNRS UMR5308, Université Claude Bernard Lyon 1, ENS de Lyon, 69365 Lyon, France; 3Institut de Génomique Fonctionnelle, Université de Montpellier, CNRS, INSERM, 34094 Montpellier, France; 4Structural and Computational Biology Unit, European Molecular Biology Laboratory (EMBL), 69117 Heidelberg, Germany; 5Department of Biochemistry and Molecular Biology, Institute for Biotechnology and Biomedicine (BIOTECMED), University of Valencia, 46010 Valencia, Spain; 6Division Phytomedicine, Thaer-Institute of Agricultural and Horticultural Sciences, Humboldt-Universität zu Berlin, Lentzeallee 55/57, 14195 Berlin, Germany

**Keywords:** Paramyxoviridae, virulence factor, overlapping genes, protein structure, viral evolution

## Abstract

The protein C is a small viral protein encoded in an overlapping frame of the P gene in the subfamily Orthoparamyxovirinae. This protein, expressed by alternative translation initiation, is a virulence factor that regulates viral transcription, replication, and production of defective interfering RNA, interferes with the host-cell innate immunity systems and supports the assembly of viral particles and budding. We expressed and purified full-length and an N-terminally truncated C protein from Tupaia paramyxovirus (TupV) C protein (genus Narmovirus). We solved the crystal structure of the C-terminal part of TupV C protein at a resolution of 2.4 Å and found that it is structurally similar to Sendai virus C protein, suggesting that despite undetectable sequence conservation, these proteins are homologous. We characterized both truncated and full-length proteins by SEC-MALLS and SEC-SAXS and described their solution structures by ensemble models. We established a mini-replicon assay for the related Nipah virus (NiV) and showed that TupV C inhibited the expression of NiV minigenome in a concentration-dependent manner as efficiently as the NiV C protein. A previous study found that the Orthoparamyxovirinae C proteins form two clusters without detectable sequence similarity, raising the question of whether they were homologous or instead had originated independently. Since TupV C and SeV C are representatives of these two clusters, our discovery that they have a similar structure indicates that all Orthoparamyxovirine C proteins are homologous. Our results also imply that, strikingly, a STAT1-binding site is encoded by exactly the same RNA region of the P/C gene across Paramyxovirinae, but in different reading frames (P or C), depending on which cluster they belong to.

## 1. Introduction

The large virus family Paramyxoviridae includes well-known human and livestock pathogenic viruses, such as measles and mumps viruses, parainfluenza viruses or rinderpest and Newcastle disease viruses [1] but also zoonotic viruses such as Hendra and Nipah viruses [2], as well as a large number of understudied viruses infecting mammals, reptiles, and fishes [3,4]. They are enveloped viruses with a negative-sense RNA genome, which are therefore classified in the phylum Negarnaviricota [5]. On the basis of the non-segmented character and organization of their genome, their replication mechanisms, and multiple sequence alignments of individual genomes or proteins, the family Paramyxoviridae has been classified in the order Mononegavirales along with the families Pneumoviridae (human respiratory syncytial virus, human metapneumovirus), Filoviridae (Ebola and Marburg viruses), and Rhabdoviridae (rabies virus) [6,7]. 

A hallmark of the family Paramyxoviridae is the architecture of the P gene, which comprises several coding regions in overlap in different reading frames [1]. The V coding region appeared to be an ancestral feature of this viral family [8], and most current paramyxoviruses produce P and V proteins by a co-transcriptional editing mechanism, in which the viral L RNA-dependent RNA polymerase (L-RdRP) stutters at a specific U-rich sequence, leading to the non-templated insertion of one or more guanosines (G) (in TupV insertion occurs at nt 2821) [9,10,11,12,13] (Figure 1). 

Members of the subfamily Orthoparamyxovirinae, which includes the genera Morbillivirus, Henipavirus, Jeilongvirus, Respirovirus, and Narmovirus, possess an additional C coding region in overlap with the 3′ end of the P coding region, and thus, the mRNAs encoding P, V, and possibly W, are polycistronic (Figure 1). In most viruses, the P coding region starts upstream of the C coding region, but in some viruses, the opposite occurs [1]. The P coding region and the C coding region are expressed by an alternative translation initiation mechanism; the ribosome fails to initiate at the first AUG start codon because of a weak Kozak sequence and initiates at the next downstream AUG codon, which happens to be in the second reading frame, leading to the expression of the other protein (Figure 1).

The C proteins of Orthoparamyxovirinae are virulence factors [14,15], which act through different mechanisms. Most C proteins have at least three functions: (1) they inhibit viral RNA synthesis, thereby controlling the rate of replication of the virus (in some viruses, they also regulate the balance between transcription and replication) and the production of defective interfering (DI) particles (by enhancing polymerase processivity) [16,17,18]; (2) they counteract antiviral innate immune responses (high degree of attenuation in KO virus) [14,19,20] and (3) they recruit cellular partners in particular, to promote viral particle assembly and budding (by recruiting ESCRT components) [21]. 

A previous sequence analysis revealed an architecture common to all Orthoparamyxoviridae C proteins, with a predicted N-terminal disordered moiety and a C-terminal helical moiety [22]. This analysis identified three groups of C proteins, represented by the measles virus, the Nipah virus, and the Sendai virus. The C proteins of the measles virus group and Nipah virus group are homologous; in contrast, the sequence and functional properties of the C proteins from the Sendai viruses were different from those of the other two groups, suggesting that the C protein might have arisen independently in the ancestor of the measles and Nipah viruses and in the ancestor of the Sendai virus group [22]. 

The C protein of Orthoparamyxovirinae, in particular of important pathogens (e.g., measles virus or Nipah virus), has resisted structure determination for over 20 years. The major bottleneck was the production of recombinant C protein, and it is known that in such cases, testing different orthologs increases the chances of solving the structure of at least one of them. Therefore, the same study tested the bacterial expression of the C protein of over 20 Orthoparamyxovirinae [22]. The most abundantly expressed C protein was that of Tupaia paramyxovirus (TupV), a virus that infects the tree shrew (Tupaia belengeri). Of interest for our study is that its P gene exhibited typical features of members of the subfamily Orthoparamyxovirinae [13] (Figure 1).

The tree shrew (Tupaia belengeri) is a small mammal that lives in a tropical rainforest in Southeast Asia and belongs to the order Scandentia. Recent genome sequencing confirmed that it is genetically closer to primates (Primates) than to rodents (Rodentia) [23], making the tree shrew a promising model for a variety of human diseases such as depression, myopia, hepatitis B and C infections, and hepatocellular carcinoma, as well as the development of therapeutic treatments, because it combines small size, easy breading, and rapid reproduction [24]. 

Here, to obtain structural information about a full-length C protein and to further investigate the evolutionary relationships between the Orthoparamyxovirinae C proteins, we expressed two constructs of the TupV C protein, we solved the crystal structure of its C-terminal folded part and characterized both the truncated and full-length proteins in solution, confirming that the N-terminal part was intrinsically disordered. Comparison with the folded domain of Sendai virus C protein revealed a common structural architecture, which suggested a descent from a common ancestor and led us to propose different possible evolution models for explaining the existence of C protein coding region in the current members of the sub-family Orthoparamyxovirinae.

## 2. Materials and Methods

### 2.1. Disorder Metaprediction 

The location of intrinsically disordered regions (IDRs) in TupV C protein was predicted by submitting its amino-acid sequence to 16 different predictors accessible through web servers and by calculating a consensus prediction as previously described [25]. The fold index was run with a window size of 51 residues [26]. For the PONDR VLXT, XL1, CAN_XT, VL3_BA, and VSL2 predictors, residues with a score > 0.5 were considered disordered [27,28,29,30]. PONDR-FIT was run using the default parameters [28]. For the IUPred2 predictors for long and short-disordered regions, residues with a score > 0.5 were considered disordered [31]. The DisEMBL (Loops/Coils, Hot-loops, and Remark465) server was run using the default parameters [32]. The GLOBPLOT server was run with the default Russel/Linding propensity [33]. The ESpritz predictors (NMR, X-ray, Disprot) were used with the default parameters [34]. HCA (Hydrophobic Cluster Analysis) plots were drawn with DRAWHCA [35].

A simple scoring procedure was then used to define consensus-disordered regions. For each prediction, residues predicted to be part of a disordered region were assigned a score of 0 while other residues were assigned a score of 1. A D-score for each amino acid was calculated by averaging the value from all predictions. Regions with a normalized D-score lower than 0.50 are considered intrinsically disordered, whereas regions with a score higher than 0.5 are either folded domains or potential molecular recognition elements (MoRE) for partner proteins [25,36,37,38,39]. A freely accessible GitHUB for automatically querying the different servers, processing the information, and calculating the D-score is now accessible at https://github.com/brisvag/dscore (accessed on 20 December 2022).

### 2.2. Protein Expression and Purification

A gene encoding full-length Tupaia paramyxovirus C protein (Uniprot: Q9WS38) in fusion with a C-terminal 3C protease cleavage site and an N-terminal 6His tag was obtained in a PopinF vector and was expressed and purified as previously described [22]. Briefly, the protein was expressed in Escherichia coli Rosetta^TM^ (DE3) cells (Novagen, Darmstadt, Germany). The bacteria were grown at 25 °C for 20 h in auto-induction ZYM5052 medium [40]. The cells were harvested by centrifugation, and the pellet was resuspended in 50 mM Tris-HCl buffer at pH 7.5 containing 300 mM NaCl, 5 mM imidazole, 50 mM arginine, 50 mM glutamate, 0.2 mM of Tris(2-carboxyethyl)phosphine (TCEP), and EDTA-free protease inhibitor cocktail (Buffer A). The cells were broken by sonication, and the soluble fraction was loaded onto a Ni^2+^ column (NiNTA, Qiagen, Hilden, Germany) equilibrated in buffer A. After washing with buffer A, the protein was eluted with 50 mM Tris-HCl buffer at pH 7.5 containing 500 mM NaCl, 250 mM imidazole, 50 mM arginine, 50 mM glutamate, and 0.2 mM of TCEP (Buffer B). The protein solution was incubated overnight at 4 °C with the 3C protease to remove the 6His tag and was concomitantly dialyzed (molecular weight cut-off of 3.5 kDa) against 50 mM Tris-HCl pH 7.5 containing 50 mM NaCl and 0.2 mM TCEP (Buffer C). The protein was then loaded on a Hitrap Q Sepharose Fast Flow column (Cytiva, Freiburg im Bresgau, Germany) equilibrated with buffer C, and then eluted with a gradient of 50 to 300 mM NaCl solution. After concentration with a centrifugal filter (molecular weight cut-off of 10 kDa—Amicon, Burlington, MA, USA), the protein was finally purified by size-exclusion chromatography (SEC) on a Hiload Superdex 75 16/600 GL (Cytiva) column equilibrated with 50 mM Tris-HCl pH 7.5, containing 150 mM NaCl and 0.2 mM TCEP (Buffer D). 

According to our bioinformatics analysis, a construct of the TupV C protein encompassing residues 54 to 153 (C_Δ53_) was subcloned between NcoI and XhoI restriction sites in a pET-M40 in fusion with TEV protease-cleavable N-terminal 6His tag and maltose binding protein (MBP) tags. The protein was expressed in E. coli BL21(DE3). The cells were grown in Luria Bertani (LB) medium at 37 °C until the optical density at 600 nm reached a value of 0.6. Protein expression was then induced by adding 0.5 mM isopropyl β-D-1-thiogalactopyranoside (IPTG), and cells were allowed to grow for 18 h at 18 °C. The cells were harvested by centrifugation, and the pellet was resuspended in buffer A. The cells were broken by sonication, and the soluble fraction was loaded onto a Ni^2+^ column (NiNTA, Qiagen) equilibrated in buffer A. After washing with 50 mM Tris-HCl buffer at pH 7.5 containing 1 M NaCl, 15 mM imidazole, 50 mM arginine, 50 mM glutamate, 0.2 mM of TCEP, and EDTA-free protease inhibitor cocktail (Buffer E), the proteins were eluted with 50 mM Tris-HCl buffer at pH 7.5 containing 300 mM NaCl, 250 mM imidazole, 50 mM arginine, 50 mM glutamate, and 0.2 mM of TCEP (Buffer F). The protein sample was incubated overnight at 4 °C with the TEV protease to remove the 6His and MBP tags. To remove all traces of cleaved MBP, the sample was applied on an amylose column equilibrated with buffer F, and the flowthrough was collected and further purified by SEC on a Superdex 75 (Cytiva) equilibrated in buffer B. Purity of the protein samples was routinely checked by SDS-PAGE and protein concentration was determined by UV spectroscopy or by refractometry during an SEC-MALLS analysis (see below).

### 2.3. Size Exclusion Chromatography Coupled with Multiangle Laser Light Scattering Experiments [41]

SEC was performed on a Superdex Increase 75 column (Cytiva) equilibrated with 50 mM Tris-HCl buffer at pH 7.5 containing 150 mM NaCl and 0.2 mM TCEP. The column was calibrated with globular proteins of known hydrodynamic radius (R_h_) [42] using commercially available calibration kits (Cytiva). Chromatographic separations were performed at room temperature with a flow rate of 0.5 mL.mn^−1^ with online MALLS detection using a DAWN-HELEOS II detector (Wyatt Technology, Santa Barbara, CA, USA), and protein concentration was measured online by differential refractive index measurements using an Optilab T-rEX detector (Wyatt Technology) and a refractive index increment, dn/dc, of 0.185 mL.g^−1^. The data were analyzed with the software ASTRA (Wyatt Technology).

### 2.4. Size Exclusion Chromatography Coupled with Small Angle X-ray Scattering Experiments [43]

Small-angle X-ray scattering (SAXS) data were collected at the BioSAXS beamline (SWING) of the Synchrotron Soleil. Samples were analyzed on a Superdex Increase 75 5/150 GL (Cytiva). The scattering from the buffer alone was measured before and after each sample measurement and used for background subtraction. All data analyses were performed using the program PRIMUS from the ATSAS package [44]. Ensembles of conformers were generated with the program RANCH, while ensemble selection was perormed with the program GAJOE [45]. 

### 2.5. X-ray Crystallography

Two datasets were collected on the PROXIMA-1 beamline at the SOLEIL Synchrotron at the wavelength of 1.07169 Å: a native crystal diffracting at 2.4 Å and a platinum derivative obtained by soaking diffracting at 2.7 Å. For both datasets, indexing and integration were performed using the XDS program suite [46]. Initial phases were obtained with the platinum derivative dataset by the single-wavelength anomalous dispersion method using the SHELX suite through the HKL2map program [47]. The alpha carbon trace from SHELXE was then used with the native dataset to build a first model with Phenix [48]. The model was improved and refined against the native dataset with COOT [49] and Refmac5 [50] from the CCP4 program suite [51]. 

### 2.6. AlphaFold Predictions

An SBGrid consortium installation of AlphaFold version 2.1.2 [52,53] was run on a local server equipped with an NVIDIA Tesla P100 GPU in order to predict the structure of NiV and MeV C proteins. The full databases were used, with max_template_date = 2022-08-29. All other parameters were left to their default values. 

### 2.7. Cell Culture

HEK293T cell line (ATCC, Manassas, VA, USA) was cultivated in Dulbecco’s Modified Eagle’s Medium (DMEM) GlutaMAX (Thermo Fisher Scientific, Waltham, MA, USA) containing 50 U/mL penicillin, 50 µg/mL streptomycin (Thermo Fisher Scientific) and 10% Fetal Calf Serum (FCS) (Eurobio, Les Ullis, France) at 37 °C and 5% CO_2_ in a humidified atmosphere. 

### 2.8. Plasmids Constructs

Nipah virus P, C, N, and L ORFs were amplified by PCR from the plasmids pTM1 NiV P, pTM1 NiV N, and pTPM1 NiV L (kind gifts of C. Basler, Mount Sinai School of Medicine, New York) [54] and cloned into the phCMV vector. To avoid the expression of Nipah C from the plasmid encoding NiV P through translational leaky scanning mechanism, the weak Kozak sequence of P (*CAT CCA*
**ATG** GAT) was replaced by the corresponding sequence of N (*ATC ATC*
**ATG** GAT), for which no expression of proteins due to leaky scanning has been reported.

The construction of the Nipah virus minigenome was performed as described for the La Crosse virus minigenome [55]. The NiV leader region, the “gene start” signal (GS) and the untranslated region from the NiV N gene were inserted upstream from the mouse RNA pol1 terminator sequence. The NiV trailer, the “gene stop” signal (GE), and the untranslated region from the NiV L gene were inserted after the RNA pol1 promoter sequence. The NiV sequence was replaced with the Renilla luciferase gene (pPol I NiV-REN), and four nucleotides (GCAT) were inserted after the stop codon of the Renilla luciferase gene in order to respect the rule of 6 [56,57]. All expression plasmids were verified by sequencing.

### 2.9. Minireplicon Assay

The subconfluent monolayer of HEK293T cells was transfected with 100 ng plasmid pPol I NiV-REN, 125 ng phCMV NiV N, 25 ng phCMV NiV P, and 350 ng phCMV NiV L using jetOPTIMUS transfection reagent (Polyplus, Illkirch-Graffenstaden, France). Additional transfection of pGL4.50 (Promega, Madison, WI, USA) encoding the firefly luciferase was performed for normalization of transfection efficiency. In all experiments, the total amount of transfected DNA was kept constant by including additional empty vector plasmid DNA where appropriate. Reporter activity was measured 24 h post-transfection using the Dual-Glo luciferase assay (Promega) following the instructions of the manufacturer. Details about the NiV mini-replicon assay will be published elsewhere.

### 2.10. TupV C Protein Interactions with the Human Proteome

For the identification of TupV C-human protein–protein interactions, DNAs encoding the C protein fused to a tandem affinity purification (TAP) tag sequence at either the amino or carboxy terminus were transfected into HEK293T cells. Briefly, 40 μg of DNA (combining at equal parts constructs with the tag at the N- and C-terminus) was added to 1.5 mL of serum-free medium (Gibco, Waltham, MA, USA) (solution A). As a control, cells were transfected with a plasmid encoding the EYFP. Separately, 1.5 mL of serum-free DMEM was supplemented with Lipofectamine 2000 (2 μL.μg^−1^ DNA; Life Technologies, Carlsbad, CA, USA) (solution B). Solutions A and B were then mixed and incubated at room temperature for at least 10 min. The mixture was then overlaid onto a 15 cm tissue culture dish. Approximately 48 h post-transfection, cells were harvested and washed with phosphate-buffered saline (PBS) (×3). Cells were then lysed with 300 μL of lysis buffer (30 mM Tris-HCl, 150 mM NaCl, 0.5% NP-40, protease inhibitors (cOmplete protease inhibitor cocktail; Roche, Basel, Switzerland), and phosphatase inhibitor (PhosSTOP; Roche), and the lysate was passed through a 30-gauge syringe (×3). The cell extract was clarified by centrifugation (10 min, 10,000× *g*) and the supernatant was transferred to a new tube. TAP, using the Strep and Flag tags in tandem, was performed as described in reference [1]. Protein digestion and identification (liquid chromatography-tandem mass spectrometry [LC-MS/MS] using an LTQ Orbitrap) was completed by the Proteomic Core Facility at the University of Valencia. Four experimental replicas were done.

The MS data were filtered according to the following selection criteria. All pseudogenes, predicted proteins, and immunoglobulin fragments were discarded. Common contaminants in MS experiments, such as keratins and trypsins, were also eliminated. The proteins purified with the EYFP control were deleted. To further eliminate common artifacts, we utilized the CRAPome database (2) of common protein contaminants. We selected proteins found with two or more peptides and in at least two experimental replicates.

## 3. Results

### 3.1. TupV C Protein Contains a Predicted Structured Region (aa 54−153)

The sequence of TupV C protein is conserved within the genus Narmovirus, except for a short, highly variable region (aa 23–34) (Figure 2A). Visual inspection of an HCA plot [35] of TupV C revealed an N-terminal part (residues 1–60) rich in charged residues and prolines, suggesting that it is globally disordered and a C-terminal part that contains five hydrophobic patches typical of α-helices (Figure 2B). In order to precisely localize the boundaries between structured and disordered regions, we predicted the location of secondary structure elements with PSIPRED [58], which predicted six helices named A to F, and combined the results of multiple predictors of intrinsic disorder into a consensus D-score, as previously described [25] (Figure 2C). In agreement with previous predictions and limited proteolysis data [22], these results indicated the presence of (1) a C-terminal region comprising four α-helices (aa 77–150) with a D-score value continuously above 0.5, and (2) another helical region (aa 54–72), with a lower D-score value (Figure 2C). 

### 3.2. Expression, Purification, and Quality Control of N-Terminally Truncated and Full-Length TupV C Proteins

Using this information, we generated and expressed in E. coli a truncated form of the C protein lacking either its 53 N-terminal residues (C_Δ53_) in addition to the full-length protein (C_FL_). We purified both proteins in two steps using affinity chromatography and size-exclusion chromatography, and we checked their purity by SDS PAGE. We determined the molecular mass and oligomerization state using SEC-MALLS. Both C_Δ53_ and C_FL_ eluted from the Superdex 75 column in a single, symmetrical peak with a constant molecular mass across the peak indicating monodisperse species (Figure 3A). The weight-average molecular mass values of 12.0 ± 0.3 kDa for C_Δ53_ and 17.8 ± 0.6 kDa for C_FL_ were in agreement with the theoretical masses of 11.9 kDa and 17.8 kDa, respectively. The polydispersity index (M_w_/M_n_) of 1.000 and 1.003, respectively, confirmed that both proteins were monodisperse and monomeric in solution. We determined the hydrodynamic radius (R_h_) values of 2.1 ± 0.1 nm for C_Δ53_ and 2.4 ± 0.1 nm for C_FL_ by calibrating the SEC column with globular proteins (Figure 3B). These values were much closer to the theoretical radii of globular proteins of the same molecular mass (1.8 and 2.1 nm, respectively) than to those of chemically unfolded proteins (3.1 and 3.8 nm, respectively) (Figure 3B). These results suggest that C_Δ53_ and C_FL_ comprise short flexible regions but are mostly compact. 

### 3.3. Crystal Structure of the Folded C-Terminal Domain Revealed Structural Similarity with the Sendai Virus C Protein

The protein C_Δ53_ crystallized readily with six molecules in the asymmetric unit in the P3_2_21 space group and diffracted at 2.4 Å resolution. The crystal structure was solved with the SAD method by soaking crystals in a platinum derivative solution, and the structure was refined to a resolution of 2.4 Å (Table 1 and Figure 3C). 

Each polypeptide chain forms four a-helices (Figure 3D): helices B, C/D (a long helix so named because it encompassed predicted helices C and D), E and F. Helix B and the N-terminal half of helix C/D are stabilized by intermolecular interactions with other C proteins (see below). Helix F is kinked at Pro127 (Figure 3D). The C-terminal part of helix C/D (aa 86–99), is packed onto a hairpin formed by helix E (aa 105–114) and helix F (aa 119–141) (Figure 3D). Helices D, E, and F form a small domain containing a hydrophobic core. Of note, the residues in the C-terminal part of the C/D helix (Y93, Q97, V99, (R/K)100, T101, L102, E108, and G109) that are strictly conserved among Narmoviruses (Figure 2A) are exposed to the solvent and could thus represent a recognition motif (Figure 3E). 

In the crystal, the proteins are packed against each other in two different modes. In the first mode (Figure 3F), two molecules of C_Δ53_ assembled head-to-tail, forming 14 interchain H-bonds, 2 salt bridges, and burying ~3300 Å^2^ of accessible surface area (PISA Complex formation Significance Score of 1.0). Helix B of each monomer binds in the fork formed by helices C/D and F of the other monomer. The interaction involves conserved residues Q63, L65, A66, L/I69, L70, and L73 in helix B, and M144 and M145 in helix F. The N-terminal part of helix C/D of two protomers packs against each other in an antiparallel orientation, mainly forming polar and symmetrical interactions. In the second mode (Figure 3G), two molecules assembled by the orthogonal interlocking of their C-terminal domain, forming 3 H-bonds and 1 salt bridge and burying ~1200 Å^2^ of accessible surface area (PISA Complex formation Significance Score of 1.0). 

The assembly of three protomers by both modes is shown in Figure 3H. The asymmetric unit comprised six molecules and had a horseshoe shape (Figure 3I). The chaining of asymmetric units leads to a helical assembly and to the formation of tubular structures with external diameters of ~9.0 nm and hexagonal internal cavities of 2.4–3.0 nm width (Figure 3J). In the crystal, the tubular helices are packed side-by-side (Figure 3K). 

A search for structural similarities with DALI [59] identified the C-terminal folded domain of the Sendai virus C protein either in complex with the N-terminal domain of human STAT1 (PDB ID: 3WWT-Z-score 4.9, 3.3 Å rmsd over 62 CA atoms), a key intermediate in the interferon signaling pathway [60], or in complex with the BRO-1 like the domain of the human protein Alix (PDB ID: 6KP3-Z-score 3.5, 2.2 Å rmsd over 54 Ca atoms), a component of the ESCRT machinery [61].

In both complexes, Sendai virus C protein (in green in Figure 4A,B) has the shape of a flat disk formed by six helices arranged in two superposed layers. In both cases, the three C-terminal helices of SeV C align with the C-terminal part of helix C/D and with helices E and F of TupV C, while additional N-terminal helices are packed on the common helical core (Figure 4C).

The interaction of SeV C with the N-terminal domain of STAT1 is mediated by residues in the first and fourth helices of C (respectively aa 104–109 and 146–161) [60]. Some of the residues involved in the interaction are conserved in TupV C (aa 98–103, Figure 4D), indicating that in principle it also has the potential to interact with STAT1 (see Section 4.3). Figure 4E presents a sequence alignment of TupV C and SeV C based on their structural superposition, made using Promals3D [62]. The second region of SeV interacting with STAT1 (aa 146–161) and its equivalent in Narmovirus C proteins are boxed (Figure 4E).

In contrast, the interaction of SeV C protein with the BRO-1 like domain of Alix, mediated by an L_122_XXW_125_ motif located in the second α-helix of SeV C, has no equivalent in the TupV C protein structure. Therefore, TupV C is not expected to bind the BRO-1-like domain of Alix, at least not in the same manner as SeV C.

### 3.4. The C Proteins of NiV, MeV, and SeV Contain a Core of 3 Helices That Superpose to Helices D-E-F of TupV C

As the C-terminal end of SeV C adopts the same fold as the core made by helices D-E-F in TupV C (Figure 5A), we wanted to verify whether the C protein of other Orthoparamyxovirinae contains the same three core helices. For this, we predicted the structure of NiV and MeV C proteins using AlphaFold2 [52,53]. The C models of NiV and MeV C contained three helices superposable with helices D-E-F of TupV C (Figure 5B,C). Thus, the helical core domain is common to the C proteins of the three groups previously identified: the measles virus group, the Nipah virus group, and the Sendai virus group [22] (Figure 5D).

We note that both NiV (166 aa) and MeV C (186 aa) proteins are longer than that of TupV (153 aa) and have a C-terminal extension, which forms one or two additional helices at the C-terminal end, respectively (Figure 5E). 

### 3.5. TupV C_Δ53_ Does Not Multimerize in Solution, Unlike in the Crystal

To determine whether TupV C_Δ53_ self-assembled in solution, we first analyzed the dependence of the elution on protein concentration by SEC-MALLS. Protein samples at a concentration ranging from 3.3 mg/mL to 23.5 mg/mL were injected into the Superdex 75 column (Figure 6). At all concentrations, the protein eluted as a single peak; its molecular mass, as determined by light scattering and refractive index, remained constant throughout the chromatographic peak (M_w_/Mn ≤ 1.005); and the weight average molecular mass remained close to the theoretical value calculated for the monomer within the experimental errors. In conclusion, TupV C_Δ53_ is essentially monomeric in solution, in contrast with its multimeric state in the crystal. Only a small variation in the elution volume might suggest the presence of larger species, which would represent only a small fraction (less than 5%).

### 3.6. In Solution, TupV C Contains Both Disordered and Globular Regions

In the crystal structure, helix B and the N-terminal half of helix C/D are stabilized by interactions between different C molecules (Figure 3). In solution, since C is monomeric (see above), we assumed that these helices might be mostly disordered. To experimentally characterize the state of these helices in C_Δ53_ and obtain information about the N-terminal disordered region of the full-length protein in solution, we performed SEC-SAXS experiments with both C_Δ53_ and C_FL_ (Table 2). We injected samples of each protein on a Superdex 75 Increase column and collected SAXS data at regular intervals along the elution peak over scattering vector (q) values ranging from 0.08 to 5.0 nm^−1^) (Figure 7A,B). 

For both proteins, the Guinier plots at low q values (q.R_g_ < 1.3) were linear, and the R_g_ values calculated by using the Guinier approximation were constant throughout the chromatographic peaks showing no dependence on protein concentration and the absence of aggregation or intermolecular interactions (Figure 7A,B). We obtained representative SAXS curves by averaging 10 frames in the center of the chromatographic peaks (Figure 7C) and determined average R_g_ values of 1.7 ± 0.1 nm for C_Δ53_ and 2.5 ± 0.1 nm for C_FL_ by using the Guinier approximation (Figure 7D) (Table 2). In the dimensionless Kratky plot for C_Δ53_, the curve reached a maximum near 1.5 for q.R_g_ values around 2.6 and rose at q.R_g_ values above 4, suggesting the presence of both a globular region and disordered regions (Figure 7E). In the plot for C_FL_, the curve reached a less pronounced maximum at 1.8 at q.R_g_ values above 2 and decreased slightly at higher q.R_g_ values before rising again at values above 6, which suggested the presence of a longer disordered region (Figure 7E). The pair distance distribution function (PDDF) revealed similar structural conclusions with D_max_ = 6.5 nm for C_Δ53_ and D_max_ = 10.5 nm for C_FL_ (Table 2). 

Comparing the experimental SAXS curve for C_Δ53_ with the theoretical SAXS curves calculated either for a monomer or for dimers extracted from the crystal structure revealed strong discrepancies (c^2^ > 100), clearly indicating that the C protein is neither a monomer nor a dimer with helix B and full helix C/D present as in the crystal (Figure 7F). This confirmed our initial intuition that the N-terminal part of the truncated protein is disordered and highly flexible in solution. 

### 3.7. In Solution, TupV C Is Well Described by Ensembles of Conformers with a Disordered N-Terminal and Fluctuating Helices B and C

To further explore the conformational diversity of C_Δ53_, we used the Ensemble Optimization Method (EOM) [45]. This approach consists of generating a large initial pool of independent conformers of the protein (using the software RANCH) and in selecting sub-ensembles of conformers that collectively reproduce the experimental SAXS curve (using the software GAJOE). 

We generated four initial ensembles of 10,000 conformers (Figure 8A): (i) **Ensemble 1**, in which folded helix B and helix C/D were conserved but the B/C loop connecting both helices was allowed to adopt random conformations, (ii) **Ensemble 2**, in which helix B and the B/C loop were allowed to adopt random conformations, while helix C/D was conserved in its folded form, (iii) **Ensemble 3**, in which helix B was conserved in its folded form while the B/C loop and helix C were allowed to adopt random conformations and (iv) **Ensemble 4**, in which helix B, the B/C loop, and helix C were allowed to adopt random conformations. 

The best fits of the experimental curve obtained with sub-ensembles of conformers taken from ensembles 1 and 2 had c^2^ values of 2.64 and 2.85, respectively. Slightly better fits were obtained with sub-ensembles of conformers taken from ensembles 3 and 4 with c^2^ values of 2.05 and 2.07, respectively. The fit obtained with a sub-ensemble model of 12 conformers selected from the initial Ensemble 4, which had the lowest c^2^ value, is shown in Figure 8B. The initial Ensemble 4 displayed Gaussian distributions of R_g_ and D_max_ values (Figure 8C,D), while selected ensembles of 12 conformers that best reproduced the SAXS curve displayed bimodal R_g_ and D_max_ distributions with a large fraction of compact conformers suggesting that the flexible N-terminal part (helix B and helix C) are somehow packed on the helical core (Figure 8C,D). 

A representative ensemble of 12 conformers is shown in Figure 8E. These results support a model in which the N-terminal part of C_Δ53_ is flexible, while the C-terminal domain is folded. However, they do not allow us to discriminate whether helix B was transiently formed or whether helix C transiently formed, extending helix D further than the region in contact with helices E and F.

We used a similar approach to explore the conformational diversity of C_FL_ with EOM. We generated four initial ensembles of 10,000 conformers with the same features as for C_Δ53_ but with an N-terminal intrinsically disordered region (NT-IDR) allowed to adopt random conformations (Figure 9A). Selections of sub-ensembles from each of these different initial ensembles yielded fits of the experimental curve of similar quality (χ^2^ = 1.25) (Figure 9B), supporting a model in which the N-terminal part of the protein was intrinsically disordered and the C-terminal part structured like in the crystal. However, these data were not sufficient to assess the fine structural organization of helices B and C/D, because the contribution of the long N-terminal disordered tail blurred the fine structure of the SAXS curve, thus preventing a more detailed determination of the fold. The initial ensembles displayed Gaussian distributions of R_g_ and D_max_ values (Figure 9C,D), while selected ensembles displayed similar bimodal R_g_ and D_max_ distributions with a large fraction of compact conformers but with a less prominent population of larger conformers than for C_Δ53_. A representative ensemble of five conformers is shown in Figure 9E.

Altogether, both the truncated (C_Δ53_) and full-length (C_FL_) proteins are well described by ensembles of conformers, in which the C-terminal structured domain is present, the N-terminal region is highly flexible but compact, and helices B and C are frayed or unfolded and/or take different orientations from those in the crystal. 

### 3.8. TupV C Protein Cross Inhibits Nipah Virus Minigenome Expression

Although the interactions between the C protein and components of the innate responses and of the ESCRT machinery vary from one virus to another, all C proteins that have been tested downregulate the production of viral RNA, in some cases demonstrating cross-species activities [18]. To test whether the TupV C protein can regulate RNA synthesis in related Orthoparamyxovirinae, we assessed whether it inhibits the expression of the minigenome of the Nipah virus. 

For this purpose, we constructed a minigenome for the Nipah virus, in which the gene of the Renilla-Luciferase was under the control of 3′ and 5′ untranslated regions of the Nipah virus (Figure 10A), and we developed a mini-replicon assay (Figure 10B). Cultured 293T cells were transfected with the plasmid encoding the minigenome together with helper plasmids encoding NiV N, P, and L-RdRP and with a plasmid encoding either NiV C protein or TupV C protein, and we monitored the expression of the minigenome by quantifying the Renilla luciferase luminescent signal (using the firefly luciferase luminescent signal as an internal reference) (Figure 10B). 

We found that both NiV and TupV C proteins inhibited the expression of Renilla luciferase in a concentration-dependent manner (Figure 10C). As a control, we assessed cell viability by measuring the firefly luminescent signal and found a slight concentration-dependent reduction in the expression of these proteins, which was accounted for in Figure 6B, indicating that, at 24 h after transfection, the reduction in Renilla luciferase expression was neither resulting from a cytopathic effect nor an overall negative effect on the cellular translation of the C protein. These results suggest that the inhibition was specific to the viral RNA synthesis machine.

### 3.9. TupV C Does Not Interact with Human STAT1 Nor with Known Interactors of Measles Virus and Nipah Virus C Proteins

Because Tupaia belengeri protein sequences are very close to the corresponding human protein sequences (98% sequence identity), we tried identifying human proteins interacting with TupV C by tandem affinity purification (TAP) combined with mass spectrometry (MS) [65]. By applying the filtering previously described [65], we identified 12 protein–protein interactions consistently picked up by this procedure (Figure 10D). Interestingly, none of these proteins were common to those previously identified by the same approach for the Nipah virus C protein [65] or for measles virus C [66]. 

In particular, no interaction was found between TupV C and human STAT1, despite the conservation between the sequence of TupV C and one of the two regions of SeV C interacting with STAT1 (Figure 4E). In principle, TupV C could only interact with T. belengeri STAT1 instead, but we think this unlikely given its extremely high sequence identity (99%) with human STAT1.

No interaction was found with Alix either, which is coherent with the fact that TupV C has no equivalent for the region of interaction between SeV C and Alix. No interaction was found either with any other component of the ESCRT machinery beyond Alix.

Although we cannot rule out false positives or false negatives in TAP experiments, the divergences between the different virus–host interactomes could reflect that each virus co-evolved with its host, and thus that different C proteins interact with different partners in the infected cells.

## 4. Discussion

### 4.1. Structural Comparisons and Predictions

Overall, the structural characterizations of the TupV C protein in the crystal and in solution confirmed its predicted architecture, with a disordered N-terminal part (NT-IDR), a C-terminal helical folded domain, and a central region whose structure depends on the context. We can state with confidence that the formation of helices B and C in the crystal is due to interactions between different C proteins, while in solution, our data suggest that these helices fluctuate, being either partially disordered or at least oriented differently than in the crystal. The comparison of the experimental SAXS curve with the theoretical curves calculated from monomeric and dimeric forms of the protein taken from the crystal clearly confirmed that the structure of the N-terminal part of C_Δ53_ in solution is different from that in the crystal. The best fit was obtained with ensembles of conformers, in which helices B and C were disordered (Figure 8).

Unfortunately, the difference in the quality of fit of the SAXS data with ensembles, in which these helices were present but allowed to move relative to the C-terminal domain, was not sufficient for clearly establishing the conformation of this region of the TupV C protein. It is possible that helices B and C are present but frayed at the ends and packed on the folded domain. In both TupV C_Δ53_ and C_FL_, the N-terminal disordered region is rather compact as seen in the R_g_ and D_max_ distribution plots of the selected ensembles (Figure 8C,D and Figure 9C,D) and in the cartoon representation of representative ensembles (Figure 8E and Figure 9E). This chain compaction could regulate some activity of the protein by hiding interacting surfaces. 

The crystal packing revealed that TupV C protein can assemble into a helical structure forming tubes, but in solution, no evidence was found for the assembly of C into dimers or higher multimers in solution. Thus, this organization could entirely be due to crystal contacts. However, the analysis by the algorithm of the webserver PISA qualifies both types of interactions with other monomers as structurally significant. Because the C proteins of various Orthoparamyxovirinae recruit components of the ESCRT machinery and participate in the budding of new viral particles [21,61], we hypothesize that C multimers form in the presence of these viral or cellular partners, in particular when the latter themselves form multimers.

### 4.2. The C Protein Originated in the Ancestor of Orthoparamyxovirinae

The Paramyxoviridae P gene is a complex transcription unit, encoding P, V, and C in overlapping reading frames (Figure 1). The appearance and evolution of the V coding region have been recently retraced [8]. In contrast, that of the C coding region has been difficult to trace, since the sequences of both P and C are poorly conserved among Paramyxoviridae, and thus beyond the reach of current phylogeny reconstruction methods. Having solved the 3D structure of TupV C protein now provides a means to retrace the origin of the C proteins since 3D structure enables the distinction of sequence homology, even in the absence of meaningful sequence similarity.

The C coding region is present in almost all members of the subfamily Orthoparamyxovirinae. A previous study [22] showed that Orthoparamyxovirinae C proteins form two main clusters according to sequence similarity: (1) the SeV group, i.e., the genera Respirovirus and Aquaparamyxovirus; and (2) the measles group and Nipah group, i.e., all other genera (Narmovirus [TupV], Morbillivirus [MeV], Henipavirus [NiV], Jeilongvirus, Salemvirus). The C protein of respiroviruses and aquaparamyxoviruses have no detectable sequence similarity with that of the other Orthoparamyxovirinae; in contrast, all other C proteins have detectable sequence similarity to each other, proving they are homologous. Likewise, the function of the C proteins differs between these two clusters: in Respirovirus, it binds STAT1 [60], whereas in other Orthoparamyxovirinae (Morbillivirus and Henipavirus), STAT1 is bound instead by the overlapping N-terminal intrinsically disordered region that is common to P, V, and W [37,67,68,69]. 

In principle, two scenarios could thus account for the present distribution and function of the C proteins: (a)The C coding region appeared in the common ancestor of Orthoparamyxovirinae and later diverged in sequence beyond recognition in respiroviruses and aquaparamyxoviruses. In this scenario, all C proteins are homologous.(b)The C coding region appeared independently in the ancestor of respiroviruses and aquaparamyxoviruses and in the rest of the Orthoparamyxovirinae. In this scenario, the C protein of respiroviruses and aquaparamyxoviruses would not be homologous to that of other Orthoparamyxovirinae.

Our discovery that the structures of respiroviruses and narmoviruses C proteins are significantly similar, and that they are also similar to AlphaFold2-predicted models of morbilliviruses and henipaviruses C proteins, indicate that the C proteins of all Orthoparamyxovirinae are homologous (scenario 1). This scenario is in complete agreement with previously noted common characteristics of the C coding regions and proteins [22]:(a)All Orthoparamyxovirinae C proteins are encoded in the same frame with respect to P (i.e., the +1 frame), and overlap at exactly the same position within the P gene (the very 5′ end);(b)All Orthoparamyxovirinae C proteins have the same architecture (proven or predicted), composed of an N-terminal disordered region and C-terminal helical domain.

### 4.3. A STAT1-binding Site Is Encoded by Exactly the Same RNA Region of the P Gene but in Different Reading Frames (P or C Coding Region) in Orthoparamyxovirinae

Many Orthoparamyxoviruses use the products of the P/V/W and C reading frames to interfere with interferon signaling, in particular by binding STAT1. In some viruses (NiV and HeV—Henipavirus and MeV—Morbillivirus), the STAT1-binding site is monopartite and located in the disordered N-terminal region of the P/V/W proteins [37,67,68,69], while in others (SeV—Respirovirus), it is bipartite, with both parts located in the ordered C-terminal domain of the C protein (thus encoded in the +1 frame with respect to P) [60]. 

Previous sequence alignments showed that the STAT1-binding site in the morbilliviruses (MeV) and henipaviruses (NiV and HeV) P/V/W proteins overlaps the segment corresponding to helix D of TupV C protein (see Figure 5 and Figure 7 in [22]). Here, structure-based alignments show that the C-terminal part of the STAT1-binding site of SeV C is similar to the same helix D of TupV C (see Section 3.3 and Figure 4D,E). Thus, our study reveals that a STAT1-binding site is encoded by exactly the same RNA region in all Orthoparamyxovirinae, but in different frames (either the P or C frame) (Figure 11). This seems too striking to be a coincidence, and suggests that this location may be particularly suitable for encoding a STAT1-binding site. 

One reason might be regulatory: both the P and C frames have the potential to be translated at the same time and same cellular location since they are encoded by the same mRNA. If the production of the P/C mRNA becomes optimized by natural selection so that the product of one of its reading frames (for example P) blocks STAT1, the product of the other frame (i.e., C) will also benefit from this regulation. One can speculate that the positioning of the STAT1 binding site in either protein may be a means of improving the fitness of these viruses and that this possibility is an additional advantage conferred by the overlap of coding regions [70].

## Figures and Tables

**Figure 1 biomolecules-13-00455-f001:**
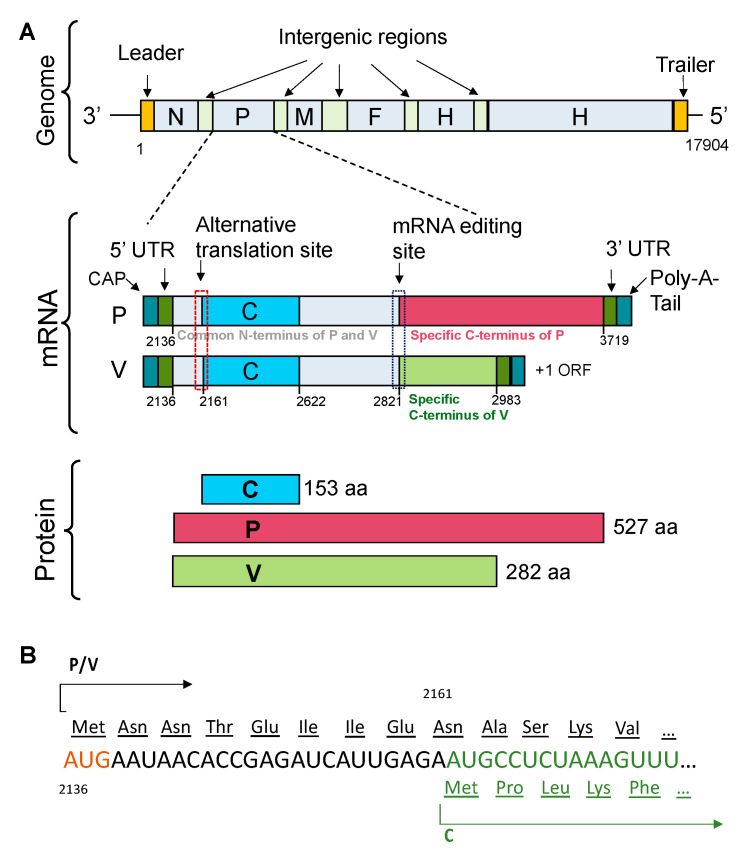
Architecture of the Tupaia paramyxovirus (TupV) genome and expression mechanisms of its P, V, and C proteins. (**A**) The upper panel shows a schematic representation of TupV genome, containing from the 3′ end to the 5′ end, the genes coding for the N, P, M, F, H, and L proteins. The middle panel shows the architecture of the mRNAs generated from the P gene by editing at the nucleotide 2821. The mRNA coding for the P protein is unedited, whereas that coding for the V protein has one G inserted at the editing site. Both mRNA contain the 3′ and 5′ UTR and are processed at both extremities with a CAP structure at the 3′ end and a polyA tail at the 5′ end. The lower panel shows the proteins expressed from these mRNAs. The P and V proteins are expressed by translation initiation at the first AUG codon of their respective mRNA, whereas the C protein is expressed by alternative translation initiation at the next downstream AUG codon from both mRNAs. (**B**) Excerpt of the mRNA coding sequence (of both P and V mRNAs) showing the first AUG initiation codon used for the translation of P or V proteins and the second AUG codon, located in the +1 frame, used for the translation for the C protein.

**Figure 2 biomolecules-13-00455-f002:**
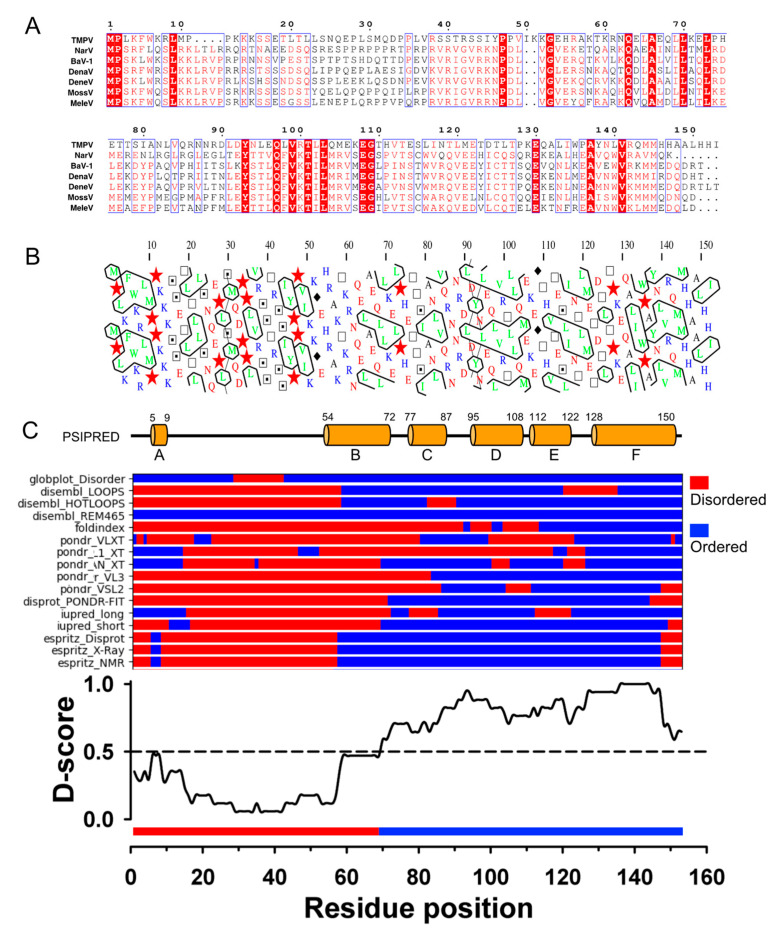
TupV C protein sequence analysis. (**A**) Multiple sequence alignment of Narmovirus C protein. Members of the Narmovirus genus and their UniProt accession number: TupV—Tupaia paramyxovirus—Q9WS38; NarV—Nariva narmovirus—B8XH61; BaV-1—Myodes narmovirus (Bank Vole virus-1)—A0A2H4PJ60; DenaV—Denalis virus—UQM99579; DeneV—Denestis virus—UQM99571.1; MossV—Q6WGM3; MeleV—Meleucus virus—UQM99623. (**B**) HCA (Hydrophobic Cluster Analysis) plot. (**C**) Secondary structure and disorder predictions. The upper part shows the secondary structure prediction from PSIPRED (the position of predicted helices is indicated by the orange cylinders with the residue numbers shown above). The middle part shows the predictions from individual predictors as obtained in the output of the D-score script. The lower part shows a consensus disordered prediction (D-score) calculated as described in [25]. The threshold to distinguish between ordered and disordered regions was set at 0.5.

**Figure 3 biomolecules-13-00455-f003:**
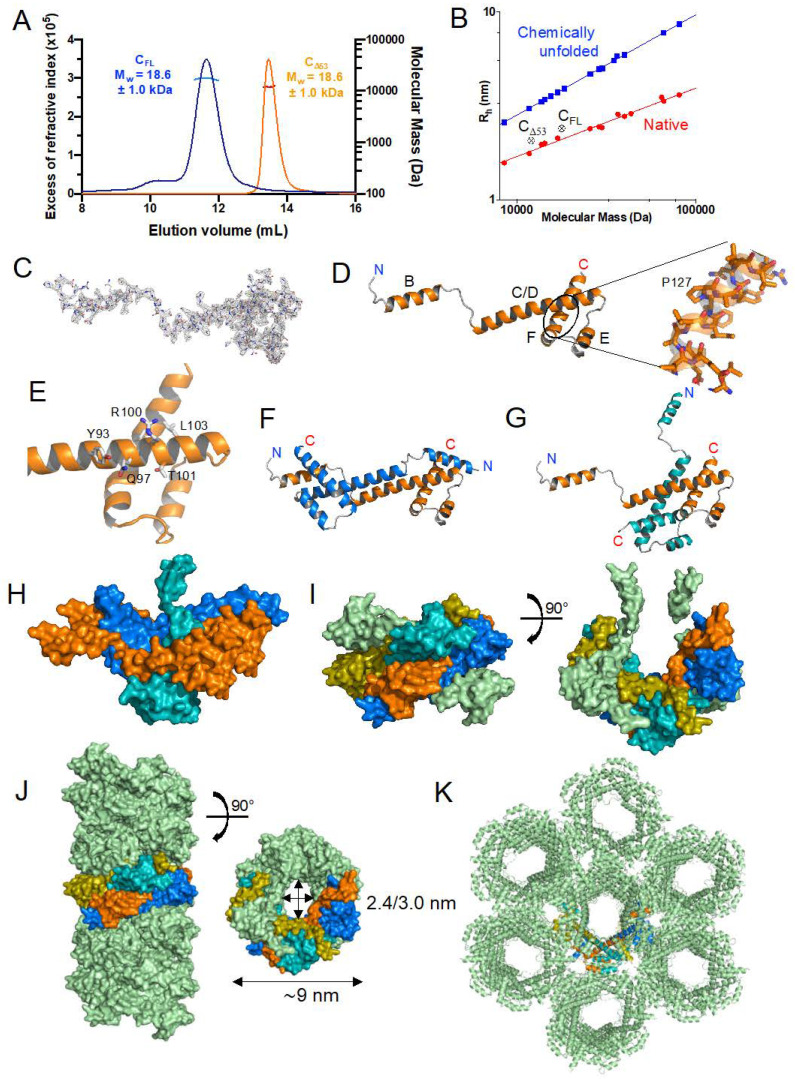
Crystal structure of TupV C protein C-terminal domain (C_Δ53_). (**A**) SEC-MALLS of TupV C_Δ53_ and C_FL_. The lines show the chromatograms monitored by differential refractive index measurements. The crosses indicate the molecular mass across each elution peak calculated from static light scattering and refractive index, and the numbers indicate the weight-averaged molecular mass (kDa) with standard deviations (the molecular mass calculated from the aa sequence are 17,807 Da for C_FL_ and 11,897 Da for C_Δ53_, respectively). (**B**) Plot of the hydrodynamic radius measured by SEC as a function of the molecular mass measured by MALLS and RI. The blue and red circles show data taken from [42] for globular proteins in native or chemically unfolded forms, respectively. (**C**) Electron density map of one protomer in the asymmetric unit contoured at 1.8 s and stick representation of the TupV C_Δ53_ protein in the crystal structure (PDB ID: 8BJW). (**D**) Cartoon representation of TupV C_Δ53_ in the crystal. The N and C-terminal residues of C_Δ53_ are indicated and helices are named B to F. (**E**) Close-up of the C-terminal domain showing residues conserved among Narmoviruses (see also Figure 2A). (**F**) Cartoon representation of the first mode of interaction between protomers within the asymmetric crystal unit. (**G**) Cartoon representation of the second mode of interaction between protomers within the asymmetric crystal unit. The orange protomer is in the same orientation as in panel (**F**). (**H**) Surface representation showing the assembly of three protomers within the asymmetric crystal unit by the two modes of interaction. The color is the same as in panel (**F**,**G**). (**I**) Surface representation in two orthogonal orientations of one asymmetric unit containing six protomers. The same three protomers of panel (**H**) are shown with the same color code, the protomer associated with the one in teal by the first mode of interaction is shown in olive, while the other two are shown in pale green. (**J**) Surface representation of the tubular assembly seen in the crystal packing. The same four protomers described in panels (**F**–**I**) are shown with the same color code, and all the others are shown in pale green. (**K**) Cartoon representation of the crystal packing showing the side-by-side assembly of the tubes described in panel (**J**) with the same four protomers using the same color code.

**Figure 4 biomolecules-13-00455-f004:**
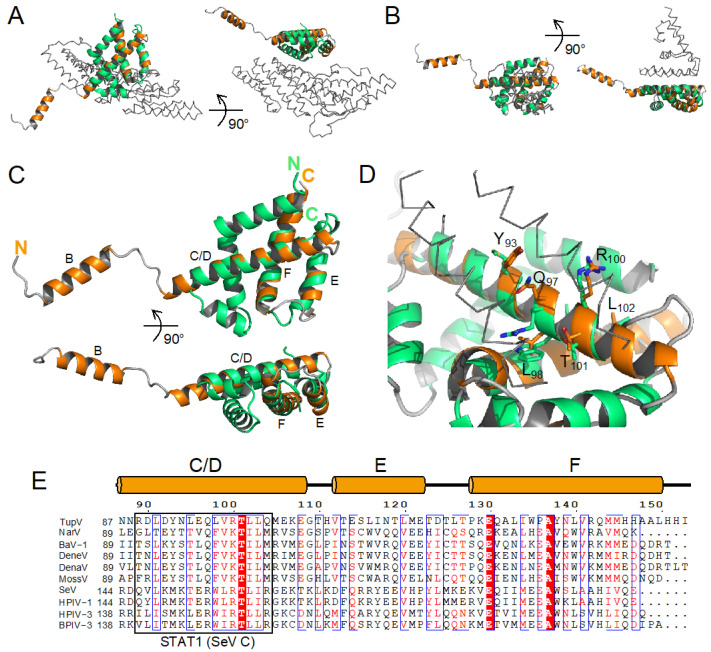
Superposition with the C-terminal domain of Sendai virus C protein. (**A**) Superposition in two orthogonal orientations of the C-terminal domain of TupV C protein with the C-terminal domain of SeV C protein complexed with the BRO-1-like domain of the protein Alix (PDB ID: 6KP3). TupV C is shown in orange, SeV C in green and the BRO-1-like domain of Alix in a grey ribbon. (**B**) Superposition in two orthogonal orientations of the C-terminal domain of TupV C protein with the C-terminal domain of SeV C protein in complex with the N-terminal domain of STAT1 (PDB ID: 3WWT). TupV C is shown in orange, SeV C in green, and STAT1 in a grey ribbon. (**C**) Superposition in two orthogonal orientations of TupV C (in orange) and SeV C (in 3WWT) taken from its complex with STAT1 (in green). (**D**) Close up of the superposition of TupV C and SeV C (in 3WWT) showing the orientation of conserved residues in both proteins. The labeled residues are those of TupV C protein. STAT1 is shown in grey ribbon. (**E**) Sequence alignment of Narmovirus and Respirovirus C proteins. Members of the Narmovirus genus and their UniProt accession number are the same as in Figure 2A, members of the Respirovirus genus are: SeV-Sendai virus-Q38KG9; HPIV1-human parainfluenza virus 1-Q8QT30; HPIV3-human parainfluenza virus 3-Q81077; BPIV1-bovine parainfluenza virus 3-P06164. The orange cylinders above the MSA indicate the location of helices in the crystal structure of TupV C. The black box indicates residues of SeV C that interact with STAT1.

**Figure 5 biomolecules-13-00455-f005:**
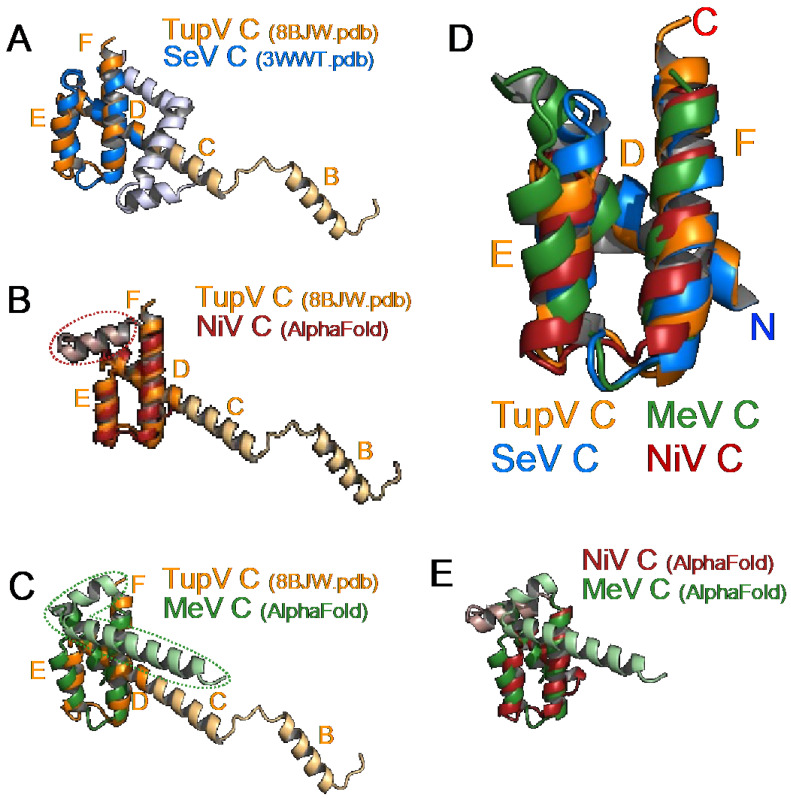
Structural alignments. (**A**) Superposition of TupV C (8BJW.pdb) and SeV C (taken from 3WWT.pdb) crystal structures. The non-superposable parts of TupV and SeV C are, respectively in light blue and light orange. (**B**) Superposition of the crystal structure of TupV C with an AlphaFold model of NiV C (aa 100–166). The averaged DDT score for this model of NiV C is 92.6, which indicates a highly reliable prediction. The additional C-terminal helix of NiV C is circled with a dotted line. (**C**) Superposition of the crystal structure of TupV C with an AlphaFold model of MeV C (aa 101–186). The averaged pLDDT score for this model is 86.0, also indicating a reliable prediction. The additional C-terminal helices of MeV C are shown and circled with a dotted line. (**D**) Superposition of (i) the helical domain of TupV C taken from the crystal structure (8BJW.pdb), (ii) the corresponding regions of SeV C taken from the crystal structure (3WWT.pdb), (iii) an AlphaFold model of MeV C and (iv) an AlphaFold model of NiV C. The N and C-termini and the name of the helices in TupV C (D–F) are labeled. (**E**) Superposition of the Alphafold models of the NiV and MeV C protein domains.

**Figure 6 biomolecules-13-00455-f006:**
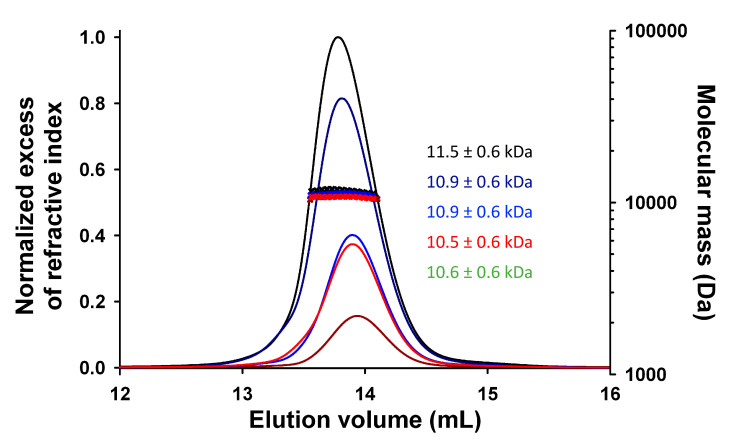
SEC MALLS experiments at different TupV C_D53_ protein concentrations. 50 mL of protein solution were injected onto a Superdex 75 column at different initial concentrations: 3.3 mg.mL^−1^ in dark red, 8.5 mg.mL^−1^ in red, 8.8 mg.mL^−1^ in blue, 20.0 mg.mL^−1^ in dark blue and 23.5 mg.mL^−1^ in black. The lines show the chromatograms monitored by refractive index and the crosses show the molecular masses calculated at each time from the light scattering intensity and refractive index. Numbers show the calculated weight average molecular masses (M_w_).

**Figure 7 biomolecules-13-00455-f007:**
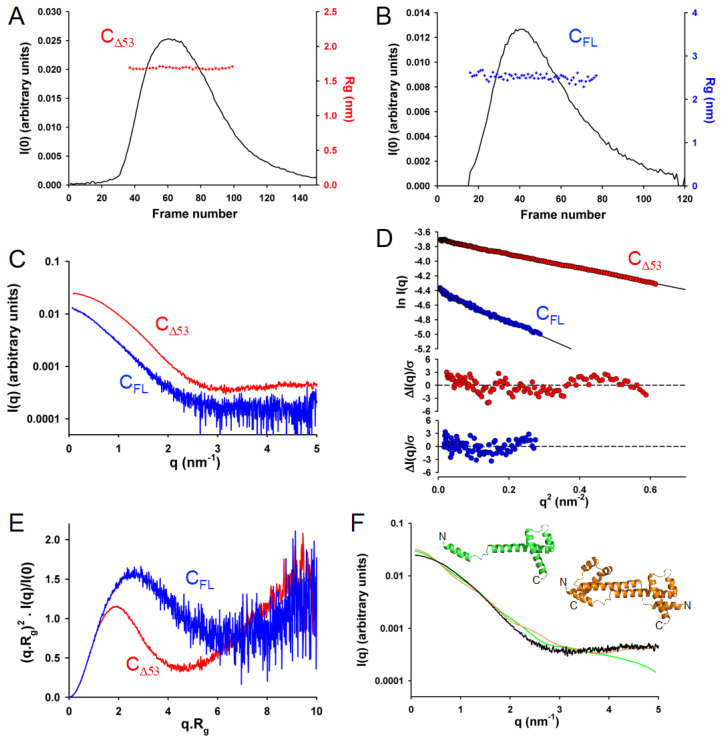
Small-angle X-ray scattering experiments. (**A**) SEC-SAXS analysis of TupV C_Δ53_ protein. The black line shows the scattering at zero angles (I(0)), which is proportional to both molecular mass and concentration, as a function of the frames recorded at regular time intervals. The red crosses show the radius of gyration calculated from the Guinier plots at different time intervals. (**B**) SEC-SAXS analysis of TupV C_FL_ protein. The black line and the blue crosses are as in panel (**A**). (**C**) Averaged scattering curves. The lines show the scattering profiles obtained for C_Δ53_ (in red) and C_FL_ (in blue) by averaging the individual profiles recorded throughout the SEC elution peak shown in panels (**A**,**B**). (**D**) Guinier plots. The upper panel shows the Guinier plots for C_Δ53_ (in red) and C_FL_ (in blue). The lower panels show the plots of the normalized residuals. (**E**) Normalized Kratky plots. (**F**) Comparison of TupV C_Δ53_ protein in solution and in crystal. The experimental SAXS curve obtained for TupV C_Δ53_ protein (black line) is compared to the theoretical curve calculated with CRYSOL for one monomer (green line) and one dimer (orange line) extracted from the crystal structure (shown in cartoon representations in green and orange, respectively). The curves were scaled in order to simply compare their shape and not to take into account differences in molecular mass.

**Figure 8 biomolecules-13-00455-f008:**
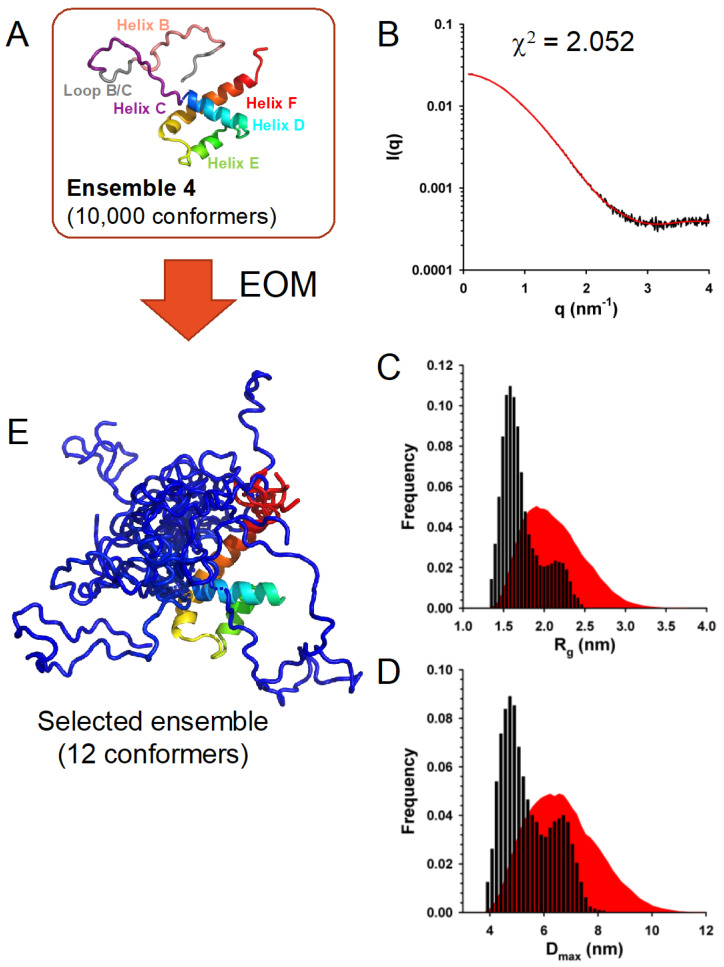
Conformational ensemble modeling of TupV C_Δ53_ protein from SAXS data. (**A**) Description of the initial ensemble, which is comprised of 10,000 conformers in which the regions corresponding to the helix B (in pink), the helix C (in purple), and the loop connecting them (in grey) were allowed to adopt random conformations. (**B**) Experimental SAXS curve for TupV C_Δ53_ (in black) and fitted curve (in red) calculated for one representative ensemble of conformers selected with the program GAJOE (shown in panel (**E**)). (**C**) Rg distribution for an ensemble of 12 conformers of TupV C_Δ53_. The red area shows the distribution for the initial ensemble of conformers. The black bars show the Rg distribution for selected ensembles of 12 conformers. (**D**) D_max_ distribution for an ensemble of 12 conformers of TupV C_Δ53_. The red area shows the distribution for the initial ensemble of conformers. The black bars show the D_max_ distribution for selected ensembles of 12 conformers. (**E**) Representative ensemble of 12 conformers that reproduced the experimental SAXS curve. The N-terminal disordered part is shown in blue and the structured domain is shown in color from blue to red going from N to C-terminal.

**Figure 9 biomolecules-13-00455-f009:**
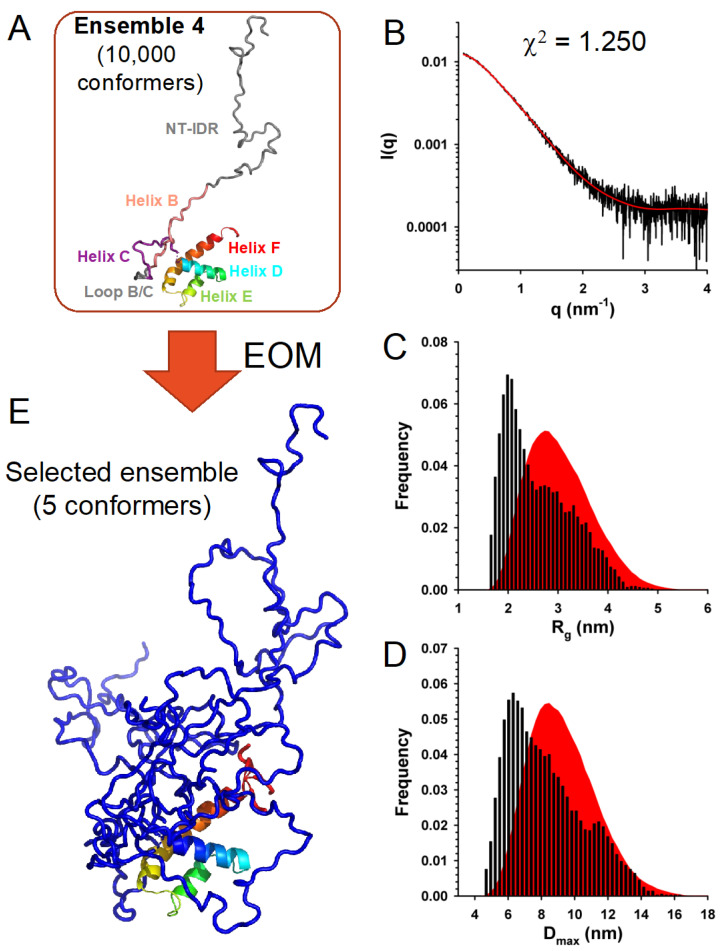
Conformational ensemble modeling of TupV C_FL_ protein from SAXS data. (**A**) Description of the initial ensemble, which is comprised of 10,000 conformers in which the regions corresponding to the N-terminal intrinsically disordered region (NT-IDR in grey), the helix B (in pink), the helix C (in purple), and the loop connecting them (in grey) were allowed to adopt random conformations. (**B**) Experimental SAXS curve for TupV C_FL_ (in black) and fitted curve (in red) calculated for one representative ensemble of conformers selected with the program GAJOE (shown in panel (**E**)). (**C**) Rg distribution for an ensemble of five conformers of TupV C_FL_. The red area shows the distribution for the initial ensemble of conformers. The black bars show the Rg distribution for selected ensembles of five conformers. (**D**) D_max_ distribution for an ensemble of five conformers of TupV C_FL_. The red area shows the distribution for the initial ensemble of conformers. The black bars show the D_max_ distribution for selected ensembles of five conformers. (**E**) Representative ensemble of five conformers that reproduced the experimental SAXS curve. The N-terminal disordered part in shown in blue and the structured domain is shown in color from blue to red going from N to C-terminal.

**Figure 10 biomolecules-13-00455-f010:**
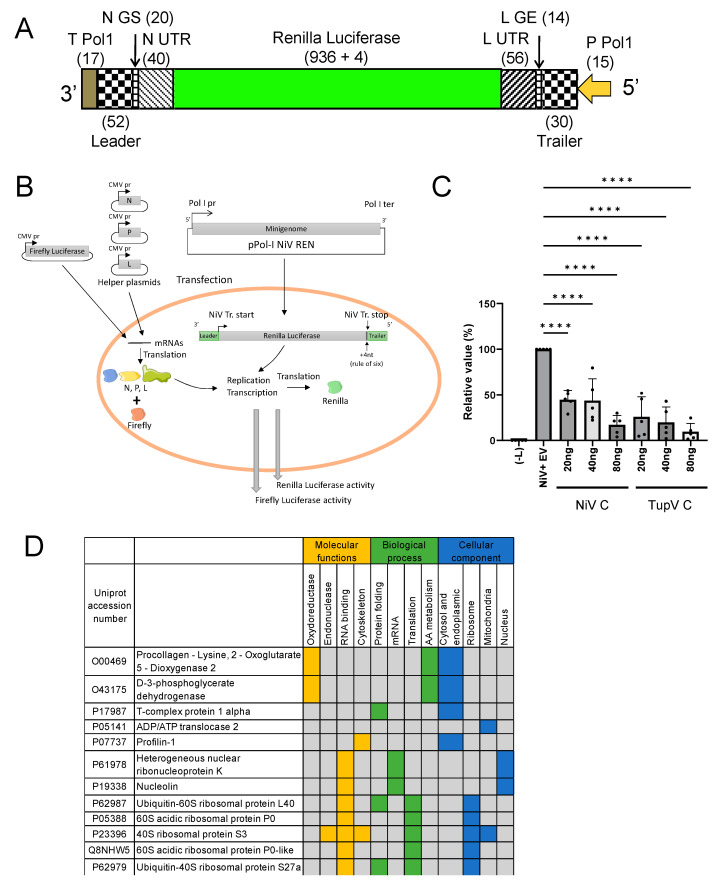
In vivo studies. (**A**) Schematic representation of the RNA minigenome. The scheme drawn as a single strand of negative-sense RNA shows the positions of the Pol1 promoter (P Pol1) and terminator (T Pol1), of the NiV leader, trailer, gene start, and untranslated regions from the N gene (N GS and N UTR), of the gene end and untranslated regions from the L gene (L GE and L UTR) and of the Renilla luciferase, including four bases added to correct the length of the minigenome to be evenly divisible by six. Lengths in nt of the different elements are shown in brackets. (**B**) Schematic representation of the mini-replicon assay. Initially, the minigenome synthesis is controlled by the polymerase I promoter in the pPol I NiV-REN plasmid. The components of the NiV ribonucleoprotein complex (N, P, L) are expressed from helper plasmids under the control of polymerase II (CMV pr). Successful encapsidation of the minigenome RNA will lead to replication of the minigenome and transcription of the reporter gene under the control of NiV polymerase L. Renilla luciferase expression is then used to quantify NiV polymerase activity. A plasmid expressing firefly luciferase under the CMV promoter (CMV pr) is used as a normalization control. (**C**) Effect of NiV and TupV C protein on NiV mini-replicon assay. HEK293T cells were transfected with plasmids encoding Nipah viral N, P, and L proteins and one NiV minigenome encoding Renilla luciferase. A plasmid encoding Firefly luciferase for normalization. In the negative control (-L), the plasmid encoding NiV L was omitted. Increasing plasmid amounts of NiV C or TupV C were co-transfected as indicated, while the amount of transfected plasmid DNA was kept constant by the addition of the pCDNA3 vector. At 24 h after transfection, luciferase activities were measured and results obtained in the absence of C protein (NiV+EV) were set to 100% whereas results of the negative control (-L) were set to 0 %. For relative Firefly expression analysis, NiV+EV values were set to 100%. Individual values are represented by dots. Mean and SD from five independent experiments are indicated. **** *p* < 0.0001; ns not significant (One-way ANOVA, Dunnett Test). (**D**) Identification of interactions between TupV C protein and human proteins by TAP-MS. A total of 12 human proteins were captured with the TupV C bait, representing different molecular functions involved in several biological processes and localized in different cellular components.

**Figure 11 biomolecules-13-00455-f011:**
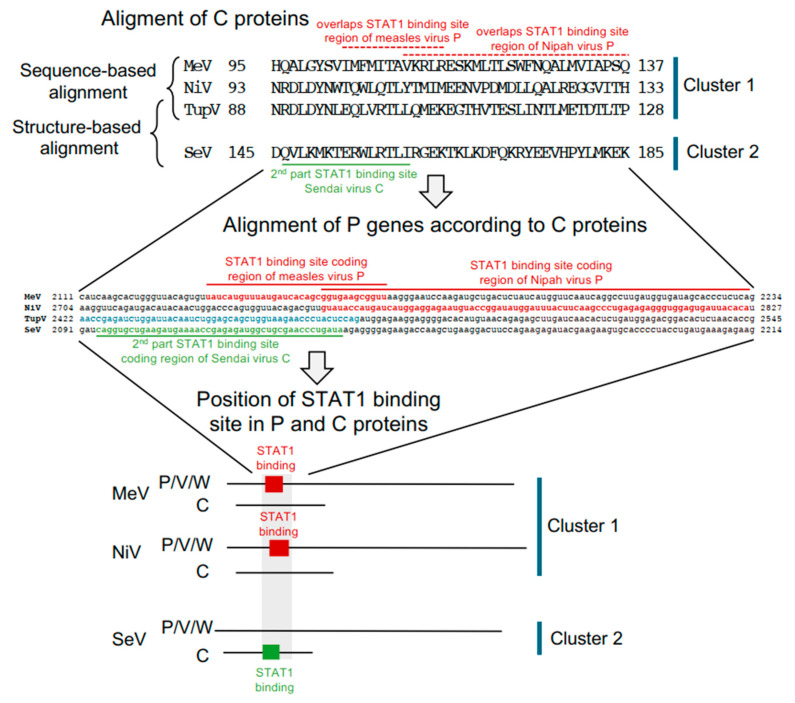
*Position of the STAT1 binding site in Orthoparamyxovirinae P or C proteins*. The upper panel of the figure shows the alignment of the part of the C proteins encompassing helices D and E of TupV C for members of both clusters. The sequence alignment of MeV, NiV, and TupV C is taken from [22], and SeV C is aligned on the TupV C sequence according to the structural alignment shown in Figure 4D. This alignment is converted into a nucleotide alignment (middle panel), where the regions coding for the STAT1 binding site are highlighted in red (encoded in the reference frame in MeV and NiV) or in green (encoded in the +1 frame in SeV). The lower panel is a schematic representation of the proteins expressed from the P gene, showing the location of the STAT1 binding site either in P/V/W proteins (cluster 1) or in C protein (cluster 2).

**Table 1 biomolecules-13-00455-t001:** Crystallographic Data Collection and Refinement Statistics.

	C_∆53_ Native	C_∆53_ Pt Derivative
Data collection		
Beamline	SOLEIL-PX1	SOLEIL-PX1
Wavelength (Å)	1.07169	1.07169
Space group	P3_1_21	P2_1_
a,b,c (Å)	135.7, 135.7, 63.1	135.6, 135.6, 64
a,b,g (°)	90, 90, 120	90, 90, 120
Resolution range (Å) ^1^	68.4–2.4	46.5–2.7
R_merge_ ^1^	0.085–0.661	0.078–2.46
I/sI ^1^	18.9–4.4	16.3–0.9
Completeness (%) ^1^	100–100	99.5–93.3
Total reflections ^1^	481,705–43,842	25,1742–17,210
Unique reflections ^1^	27,136–2824	18,927–1316
Multiplicity ^1^	17.8–15.5	
Anomalous completeness (%) ^1^	n/a	99.3–91.7
Anomalous multiplicity ^1^	n/a	6.9–6.7
DelAnom correlation ^1^	n/a	0.432–0.042
Refinement		n/a
R_work_/R_free_	0.22 / 0.25	
No. water	49	
R.m.s. deviations		
Bond length (Å)	0.0099	
Bond angle (°)	1.5851	
Ramachadran statistic ^2^		n/a
Favoured (%)	98.35	
Allowed (%)	1.47	
Outlier (%)	0.18	
PDB code	8BJW	n/a

^1^ Values in for all data set and the high-resolution shell; ^2^ Values from MolProbity webserver.

**Table 2 biomolecules-13-00455-t002:** Small-angle X-ray Scattering Data Collection and Analysis.

	C_FL_	C_Δ53_
Data collection parameters		
*Instrument*	SOLEIL-SWING	
*Energy* (keV)	12.000	
*Detector*	EigerX-4M	
*Detector distance* (m)	2	
*Exposure* (s per image)	1	
*q range* (Å^−1^)	0.008–0.50	
*Column*	S75inc 5/150 GL	
*Flow rate* (mL.min^−1^)	0.3	
*Sample concentrations* (mg.mL^−1^)		
*Injection volume* (μL)	50	50
*Temperature* (K)	293	293
Guinier Analysis		
*I(0)* (*A.U.*)	1.2·10^−2^ ± 0.03·10^−2^	2.4·10^−2^ ± 0.01·10^−2^
*Rg* (nm)	2.5 ± 0.1	1.7 ± 0.1
*qrgmin*	0.2951	0.2702
*qrgmax*	1.3070	1.3027
P(R) analysis ^1^		
*Dmax* (nm)	10.5	6.4
Molecular weight ^2^		
*Theoretical Mw* (kDa) ^3^	17,807	11,897
*Measured Mw* (kDa)	18.6 ± 0.8	12.3 ± 0.4

^1^ Calculated with GNOM [63] and BIFT [64]. ^2^ Calculated from the volume of correlation. ^3^ Calculated from the aa sequence.

## Data Availability

Coordinates and structure factors have been deposited in the Protein Data Bank under accession codes 8BJW.pdb.

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
