# Peer review of "Orthoparamyxovirinae C Proteins Have a Common Origin and a Common Structural Organization"

_biomolecules, 2023, doi:10.3390/biom13030455_

Round 1

Reviewer 1 Report

This paper by Roy et al., reports original experimental and informatic studies of the C protein from the Tupaia paramyxovirus (TupV). Based on experience with other C proteins and bioinformatics predictors, the authors suspected a disordered N-terminal region and folded C-terminal domain. They expressed 2 variants, full length and a C-terminal domain in bacteria, crystalized the c-terminal fragment, compared these two proteins in solution with several methods including size-exclusion chromatography, multi-angle light scattering, SAXS, Mass-spec, AlphaFold calculations, and live-cell molecular-interaction assays.

They find that the N-terminal domain is disordered but compact and that there are several alpha helices in the c-terminal domain that should homology with other C proteins from related viruses. Some helix contacts are stabilized in the crystal structure but are missing in solution. Computational approaches are used to estimate the ensemble of conformations achieved in solution where the B and C helices are proposed to fluctuate. Comparisons with alphafold predictions for other C proteins from other viruses show conserved features of the c-terminal domain folds. The live cell assays show the TupV C protein inhibits viral replication similar to NiV C protein, but does not interact with human STAT1 or other targets of other C proteins tested.  Finally, the paper concludes with arguments about the evolutionary relationships of the C proteins being homologous in all Orthoparamyxovirinae.

This is a nice demonstration of integrative structural biology approaches to reveal the structural ensemble of the C protein from TupV. I am not an expert in x-ray crystallography or some of the other methods, but the experiments appear to be done at the standards of the field. The comparisons between the proteins of the different virus families creates a compelling story that is interesting from a virology standpoint. For these reasons, readers will appreciate this paper and I recommend publication.

I have only one issue in the paper that the authors might consider. I did not fully understand the point the authors were trying to make at the end of the introduction in the 2 paragraphs between lines 100 and 113. I believe they were describing historical experimental challenges in this field of study, but perhaps more explicit discussion would help non-experts.

A couple very minor typos are:

Line 32 – former should be form

Line 249 – missing period

Line 302 – missing a closing parenthesis

Author Response

Thank you to the reviewer for pointing out the difficulty for understanding one paragraph of the introduction.

Referee #1

I have only one issue in the paper that the authors might consider. I did not fully understand the point the authors were trying to make at the end of the introduction in the 2 paragraphs between lines 100 and 113. I believe they were describing historical experimental challenges in this field of study, but perhaps more explicit discussion would help non-experts.

We have clarified lines 100-113 by better explaining the rationale for the choice of Tupaia paramyxovirus C protein, and ensuring a better logical link between the two paragraphs. 

A couple very minor typos are:

Line 32 – former should be form

Corrected

Line 249 – missing period

Corrected

Line 302 – missing a closing parenthesis

Corrected

Reviewer 2 Report

Review of the manuscript entitled “Orthoparamyxovirinae C proteins have a common origin and a common structural organization (#2186301).

            The authors expressed and purified full-length and an N-terminally truncated C protein from Tupaia paramyxovirus (TupV) C protein to analyze the evolutionary relationships between the Orthoparamyxovirinae C proteins. They solved the crystal structure of the truncated Tupaia C protein and observed similarities with the Sendai C protein despite the absence of similarity in their genome coding sequences. They found common structural architecture with the Sendai C protein, suggesting a common ancestor for them. They predicted the potential STAT1 interaction domain in the Tupaia C protein comparing the structures of both proteins and identified 12 interacted-cellular proteins. The authors deal with a topic of interest in a well written manuscript.

I find the work able to be published in Biomolecules, answering questions below that I am sure would improve the manuscript.

1.- Why have the authors used different protein (full length and truncated C proteins) purifications methods?

2.- Line 232. Please, change the dots for / in “50 U.ML and ug.mL”. Please, proceed in the same manner in line 465.

3.- Lines 241 and 260. Please detail this construct or include the corresponding reference.

4.- A scheme of the plasmids used (pPol I NiV-REN, phCMV NiV N, phCMV NiV P, phCMV NiV L) would be grateful, even as a supplementary figure.

5.- In the Figure 5, the D is duplicated and the C is missed. Please, correct it.

6.- Line 664. Please revise the “(A)” that appears in that line.

7.- How do the authors explain the lack of coincidence between the 12 proteins that they identify as those that interact with Tupaia C protein and those that have been identified for Measles and Nipah?

8.- Could the authors comment their thoughts about the differences observed in the structures in solution and in the crystal?

Author Response

We thank the reviewer for raising different issues, which we have addressed as follow:

1.- Why have the authors used different protein (full length and truncated C proteins) purifications methods?

 The constructs were generated by different people and the choice of different strategies was serendipitous and not done in purpose or because of problems

2.- Line 232. Please, change the dots for / in “50 U.ML and ug.mL”. Please, proceed in the same manner in line 465.

 Corrected

3.- Lines 241 and 260. Please detail this construct or include the corresponding reference.

Line 241 : “The expression of NiV C from the P gene was reduced by replacing the Kozak sequence of the P protein by that of the N protein (a detailed description of this construct will be published elsewhere).”

Was replaced by

“To avoid expression of Nipah C from the plasmid encoding NiV P through translational leaky scanning mechanism, the weak Kozak sequence of P (CAT CCA ATG GAT) was replaced by the corresponding sequence of N (ATC ATC ATG GAT), for which no expression of proteins due to leaky scanning has been reported..”

4.- A scheme of the plasmids used (pPol I NiV-REN, phCMV NiV N, phCMV NiV P, phCMV NiV L) would be grateful, even as a supplementary figure.

An additional panel was added in Figure 10 that describe the minireplicon assay including some description of the different plasmids

5.- In the Figure 5, the D is duplicated and the C is missed. Please, correct it.

  Corrected

6.- Line 664. Please revise the “(A)” that appears in that line.

 Corrected

7.- How do the authors explain the lack of coincidence between the 12 proteins that they identify as those that interact with Tupaia C protein and those that have been identified for Measles and Nipah?

We have no good explanation for this. We hypothesize that all C proteins have recurrent functions that were conserved through evolution but also that each one has evolved new functions in the context of the infected host.

We have added a sentence (line 694-696  of the revised version) to comment on this.

8.- Could the authors comment their thoughts about the differences observed in the structures in solution and in the crystal?

It has been observed in different crystal structures that a protein region that is intrinsically disordered or exchanging between different conformations can be stabilized by contacts with neighbors in the crystal and will thus appear folded. We stated this in the discussion:

“Overall, the structural characterizations of the TupV C protein in the crystal and in solution confirmed its predicted architecture, with a disordered N-terminal part (NT-IDR), a C-terminal helical folded domain, and a central region whose structure depends on the context. We can state with confidence that the formation of helices B and C in the crystal is due to interactions between different C proteins, while in solution, our data suggest that these helices fluctuate, being either partially disordered or at least oriented differently than in the crystal.

We modified a sentence to clarify the fact that only the N-terminal part of our construct was different in solution and in the crystal.